# On a Spurious Interaction between Uncertainty Scores & Answer Evaluation Metrics in Generative QA Tasks

**Andrea Santilli**[*]
Sapienza University of Rome

**Miao Xiong**[*]
National University of Singapore

**Michael Kirchhof**[*]
University of Tübingen

**Pau Rodriguez**
Apple

**Federico Danieli**
Apple

**Xavier Suau**
Apple

**Luca Zappella**
Apple

**Sinead Williamson**
Apple

**Adam Goliński**
Apple

## Abstract

Knowing when a language model is uncertain about its generations is a key challenge for enhancing LLMs' safety and reliability. An increasing issue in the field of Uncertainty Quantification (UQ) for Large Language Models (LLMs) is that the performance values reported across papers are often incomparable, and sometimes even conflicting, due to different evaluation protocols. In this paper, we highlight that some UQ methods and answer evaluation metrics are spuriously correlated via the response length, which leads to falsely elevated performances of uncertainty scores that are sensitive to response length, such as sequence probability. We perform empirical evaluations according to two different protocols in the related literature, one using a substring-overlap-based evaluation metric, and one using an LLM-as-a-judge approach, and show that the conflicting conclusions between these two works can be attributed to this interaction.

## 1 Introduction

Large Language Models (LLMs) have recently grown in popularity for their strong general-purpose performance in Natural Language Generation (NLG) [34, 7, 30, 14]. However, since they are statistical models trained to approximate the distribution of the training data, their answers sometimes contain errors, commonly dubbed "hallucinations" [3, 13]. These errors can have serious consequences as LLMs are increasingly being applied in critical real-world domains like medicine [4, 28, 25, 33, 32]. Recent theoretical research raises the concern that LLM errors might even be inevitably tied to the nature of those models and their training methodology, and hence unavoidable, even in the long term [2, 17, 35]. This makes the development of methods that quantify the certainty about models' output and allow for the detection of their errors a paramount and lasting priority.

However, an increasing issue in the fast-developing field of LLM uncertainty quantification (UQ) is that the performance values reported across papers are often incomparable and sometimes even directly conflicting. For example, semantic entropy [10], an approach based on sampling multiple answers and semantically clustering them to detect which probability the LLM assigns to its answer, claims to outperform the predictive entropy baseline. Accuracy probes are claimed to outperform both the semantic-clustering-based and logit baselines [18]. However, [19] claims that semantic entropy outperforms accuracy probes, as well as the logit baselines. The only reoccuring finding these papers

---

[*]Work done during an internship at Apple.

Published at NeurIPS Safe Generative AI Workshop 2024

agree on seems to be that new, complicated methods outperform the simple logit baselines—except that the recent LM-polygraph benchmark [9] reports that simple logit baselines are surprisingly competitive with complicated methods, including semantic entropy. The shape of the performance landscape of these methods is strongly contested.

In this paper, we exemplarily focus on the contradicting results between [9] and [10]: in the former, no clear advantage is observed among the UQ methods, with simple logit-based methods proving highly competitive, while in the latter, semantic entropy emerges as the most effective approach. We find that, holding all other design choices constant, a fundamental sub-component of the evaluation protocol that drives most of the discrepancy is the *answer evaluation metric*, i.e., the numerical score that judges how well an LLM answer matches the reference answer. If the answer evaluation metric is based on a substring-overlap metric, such as Rouge-L [22] or SQuAD [31], or a fixed-length learned representation comparison metric like BERTScore [36], then certain logit-based UQ methods are heavily favored. Conversely, if using an LLM itself to judge how well the two answers match, results are more conservative, and no stark advantage is observed among the methods. Which set of results should we trust? To this end, we ask multiple annotators to manually label the correctness of the LLM's responses w.r.t. the reference answers and compare them to the considered answer evaluation metrics. In line with prior literature [38], we find that the LM-as-a-judge approach is more robust. Our key contribution is to show that there exists a correlation between the response length and the value of some evaluation metrics. Even after binarization via thresholding using the commonly used threshold values, those metrics have higher False-Negatives when the model responses are long. This spuriously favors the uncertainty scores sensitive to the response length, such as the sequence probability since longer responses tend to have lower probability.

The paper is structured as follows: in Section 2, we outline elements of the design of the evaluation protocol for selective answering in NLG QA tasks, particularly the answer evaluation metric. In Section 3, we focus on the discrepancy between the benchmarking conclusions of [9] and [10], and in Section 4, we evaluate the reliability of the compared answer evaluation metrics. In Section 5, we uncover a non-obvious interaction between the answer evaluation metric and the response length.

## 2 Evaluating uncertainty for selective answering in NLG QA tasks

Estimating the uncertainty of an LLM answer is a function of both the model and the task being solved. This complexity prevents tackling the task of uncertainty quantification like a standard supervised learning tasks which aim to approximate a functional mapping from inputs to outputs. Instead of relying on the labeled input-output pairs, the performance of uncertainty quantification is assessed through its impact on downstream tasks, such as selective prediction (or *selective answering* in the NLG context) [8, 12], out-of-distribution detection [27], or more complex decision-making under uncertainty problem settings [24].

In this paper, in line with previous related works [23, 9, 10, 1], we focus on the evaluation in the selective prediction setting [8, 12]. The goal of selective prediction is to allow the ML system to abstain from answering the questions/prompts it is uncertain about. In particular, we consider selective answering of Question-Answering (QA) tasks. We focus on QA tasks because i) they are the standard choice in the UQ literature, and ii) each question has a single, well-defined answer, making them ideal for defining a correctness function.

Even for tasks as simple as QA tasks, there exist many design and implementation choices one needs to make that might lead to discrepancies in the evaluation outcomes and hence the conclusions drawn. In this work, we focus primarily on one of them—the answer evaluation metric.

**Answer evaluation metric.** In order to judge the performance of the system, we need a way to judge whether the LLM's response answers the question correctly or not. In general, considering the variety of NLG tasks, such *answer evaluation metric* yield a score that is either continuous (e.g., in case of summarization or translation quality) or binary (like in the case of QA tasks, where the notion of correctness of the answer is binary). For QA datasets, typically each question $x$ has an associated reference answer $y$. The answer evaluation metric $\ell(\cdot, \cdot)$ compares a particular model response $\hat{y}$ to a single reference answer $y$ to determine whether the free-form answer of the LLM matches the reference answer from the dataset, $\ell(\hat{y}, y)$. There are multiple design-choices here: a) Rouge-L [22], a substring matching criterion, used in [9, 20], that computes the F1-score of the Longest Common Subsequence (LCS) between the reference answer and the generated answer (longest sequence of

words appearing in both texts); b) SQuAD [31], a similar substring matching criterion, used in [10]; c) the BERTScore [36] that compares the cosine-distance of the BERT embeddings [6] of both answers, used in [9]; d) LM-as-a-judge [37] (also known as LM-grader) which uses strong LLMs to decide whether the model response is semantically equivalent to the reference answer in the context of the question, which is used with varying models and prompt formats [10, 23]. As we can see, there are several choices that determine *correctness* that we study in this work, becoming an important design choice that will impact evaluation results.

**UQ performance metrics.** Computing the UQ performance metric involves comparing the correctness value to the uncertainty estimate. In this way, it's possible to measure whether higher uncertainty estimates are actually indicative of incorrectness. For example, the area under the Receiver-Operator-Characteristic curve (AUROC) is a popular metric for quantifying the performance of UQ methods against binary correctness values.

**QA datasets.** There exists a large variety of QA datasets in the NLP literature. In this work, we consider the ones most commonly used for UQ: TriviaQA [15] tests trivia facts, SQuAD [31] general knowledge questions that are answerable given the context, and NQ-Open [21] comprises natural Google queries, and SVAMP [29] consisting of math text questions.

**NLG UQ methods.** There are many approaches that output (un)certainty estimates that can be used to predict the (in)correctness of LLM outputs. We can group these approaches into three main categories: 1) *Single-sample methods*: methods that require a single forward pass from the model and that generally use directly the logits and probability distributions over the vocabulary space provided as output from the model; 2) *Multiple-sample methods*: methods that, given a prompt $x$, sample multiple possible outputs for the same prompt and compute an uncertainty score based on these outputs; 3) *Learned methods*: usually probes or small networks directly trained to predict the accuracy of the model given the prompt and the answer. We provide a more detailed description of the methods we evaluate in App. C.

## 3   Ablating the impact of answer evaluation metric on evaluation conclusions

In this section, we ablate the impact of the answer evaluation metric on the conclusions of the evaluation of the performance of various UQ methods—we compare two binary metrics: a) Rouge-L score thresholded at 0.5 (following [5, 10]), and b) LM-as-a-grader with Llama3-8b-instruct (see App. A for details). Our qualitative results for these two metrics align with the findings of [9] and [10], respectively. The Rouge-L-based results of [9] suggest that the *Maximum Sequence Probability* (MSP, Eq. 1 applied to the greedy decoding) is almost consistently the best performing UQ estimator for the QA tasks considered in their benchmark. In contrast, although [10] does not evaluate the performance of *Sequence Probability*, their LM-as-a-grader-based results indicate that *Semantic Entropy* (Eq. 5) variants generally outperform the related logit-based *Naive Entropy* baseline (Eq. 4). Here, we show this discrepancy can be largely attributed to the choice of the answer evaluation metric. In the next sections, we seek to explain this phenomena in more detail.

We base the experimental setup on the open-source codebase of [10], extending it with the UQ estimators in App. C. We follow the evaluation protocol of [10] with minor changes for practical reasons, see App. B for details. We use the long-form setting (sentence-length generations) of [10] which is similar to [9]. Note that our goal is not to exactly replicate the results of [9, 10], but to ablate the impact of the choice of the answer evaluation metric on the qualitative conclusions. We evaluate performance across four QA datasets (TriviaQA [15], SQuAD [31], NQ-Open [21], SVAMP [29]), and report the average performance across datasets and the 95% confidence intervals estimated via bootstrapping for each dataset and averaging across the datasets.

Comparing the results in Figures 1a and 1b, the most stark difference is the relative performance of *Sequence Probability* and *P(IK)* w.r.t. the other methods. For the Rouge-L metric, these two methods make for the top 2 methods for six of the seven evaluated models (and are within the confidence intervals of the top-2 methods for the last, Llama2-7b). In contrast, for the LLM-as-a-judge metric, only one of them ever reaches the top-4 rank, namely only on Falcon-7b. These conclusions individually align with those of [9] and [10], but, as noted above, indeed contradict each other. Let us now shed a light on what the disagreement arises from.

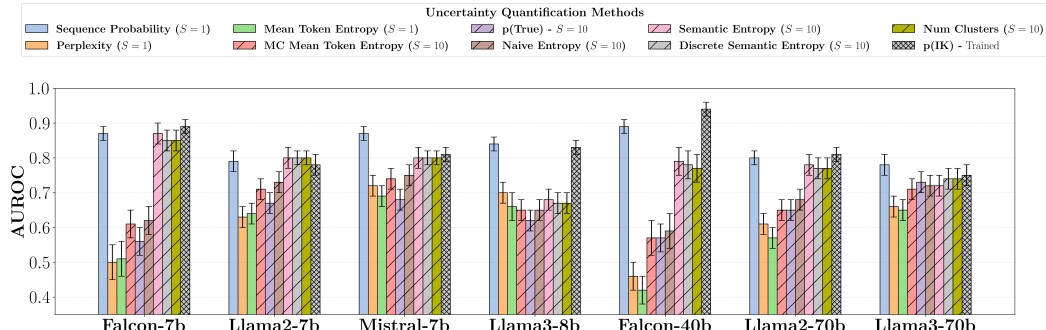

(a) Answer evaluation metric: binary score of Rouge-L score thresholded at 0.5.

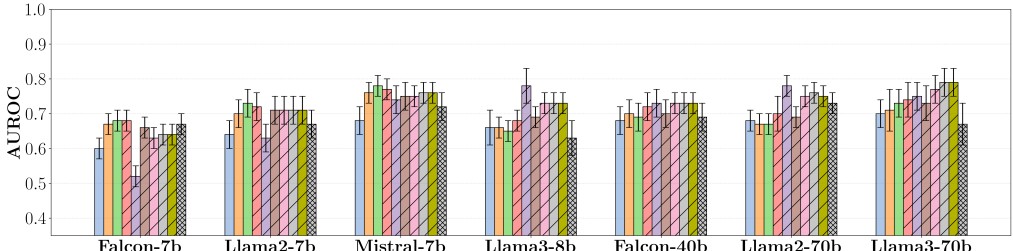

(b) Answer evaluation metric: binary score with LM-as-a-grader with llama-3-8b-instruct.

Figure 1: Comparison of results in the long-form setting, averaged over 4 datasets. 95% bootstrap confidence intervals over the means. The dashed bars are the multi-sample methods. The crossed bar is a learned method. Details in the main text.

|  | Rouge-L > 0.5 | Human 1 | Human 2 | Human 3 |
|---|---|---|---|---|
| Llama3-8b-instruct | 0.71 | 0.91 | 0.93 | 0.94 |
| Rouge-L > 0.5 |  | 0.68 | 0.69 | 0.71 |
| Human 1 |  |  | 0.97 | 0.95 |
| Human 2 |  |  |  | 0.95 |

Table 1: Agreement rates between human annotators and answer evaluation metrics evaluated on 150 questions from the TriviaQA dataset, one low-temperature $T = 0.1$ sample for each question.

# 4 Evaluating the answer evaluation metrics

In the previous section, we saw that the choice of the answer evaluation metric can have a significant impact on the conclusions from benchmarking the UQ methods. So which metric's results should we trust more? In this section, we evaluate the performance of several answer evaluation metrics w.r.t. human annotators labels.

We had 3 human annotators label the binary correctness of the model response w.r.t. the reference answer for the samples from Llama2-7b-chat on 150 questions from the TriviaQA dataset, one low-temperature $T = 0.1$ sample for each question, as per the protocol of Section 3. We report the results in Table 1. First, the inter-human agreement is over 95%, showing that their ratings are relatively noise-free. LM-as-a-grader with Llama3-8b-instruct has an agreement of over 91% with each of the human labelers, while the agreement between Rouge-L-metric and the human labelers hovers around 70%. These results suggest that using the LM-as-a-grader is a more trustworthy approach. This is in line with the literature [38]. While the results in this section are not novel, they are a stepping stone to the observation of the next section.

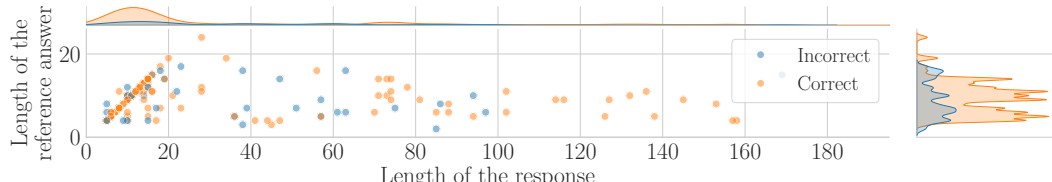

Figure 2: Scatter plot of the length of the reference answer versus the length of the sampled model response on the 150 manually annotated TriviaQA questions. Correctness labels by human annotators.

## 5 A spurious interaction between answer evaluation metrics and UQ methods

In the previous section, we concluded that LM-as-a-grader metric is more reliable than the thresholded Rouge-L metric. However, by itself, that does not explain the change in the relative performance of various UQ methods we saw in Section 3. After all, if the errors of the Rouge-L-based answer evaluation metric were made at random, we would not expect that effect. There must be a systematic effect driving that behavior.

In this section, we show that the reason for the elevated performance of *Sequence Probability* score for the Rouge-L-based metric is a spurious interaction with some answer evaluation metrics, due to both the UQ score and the answer-evaluation-metric being correlated with the model's response length. First, we consider the case of a continuous Rouge-L score (as per [9]), and then the case when the score is binarized through thresholding. Finally, we show that the same effect applies to two other answer evaluation metrics used in the literature, SQuAD [10] and BERTScore [9].

**Continuous score case.**    Figure 2 shows that the reference answers are consistently rather short compared to the broader distribution of the LLM answers. The distribution of the LLM responses' length has a long tail. And even the responses that are multiple times longer than the reference answer are often correct, as judged by human annotators. But here is where the discrepancy of the Rouge-L-score becomes apparent: For two responses that are equally correct according to human labelers (and LM-as-a-grader), the continuous Rouge-L score assigns a higher evaluation score to the shorter response, see Figure 3a. At the same time, *Sequence Probability* assigns lower uncertainty to shorter responses because every term in the product of Eq. 1 is smaller than 1. The two quantities are spuriously correlated via the response length, which leads to a spuriously inflated estimate of UQ performance of *Sequence Probability* when judged by Rouge-L. In the case of [9], this effect leads to the conclusion that it is one of the top-performing UQ scores.

**Binarized score case.**    The binarization of the Rouge-L score via thresholding has the potential to alleviate this spurious effect. When binarizing, we break the direct relationship between the resulting binary score and the response length. However, the binarized score still carries a relationship to the response length. Figure 3b shows the proportion of False Negatives, the responses that were deemed incorrect according to the Rouge-L metric binarized at 0.5 (a popular thresholding value [5]) but were deemed correct by all three human labelers. Still, the proportion of False Negatives grows with the length of the response. Like in the case of using continuous Rouge-L score, this behavior leads to a spurious correlation with the Sequence Probability score, and explains the effect from Section 3. In Figures 4 and 5, we show the same qualitative effect holds for SQuAD and BERTScore metrics.

**Discussion.**    This analysis highlights a non-trivial confounding between the evaluated uncertainty method and the answer evaluation metric, which is different and separate from the observation of Section 4 and prior literature [38] on the fact that LM-as-a-judge approach achieves better agreement with human labelers than other answer evaluation metrics. Pointing to this effect as the source of the performance difference of *P(IK)* in Section 3 is not as straightforward, but we speculate that it is feasible for the learned UQ approaches to use the signal about the response length to take advantage of this spurious interaction. As a practical take-away, we recommend practitioners to use LM-as-a-judge metrics where possible. One further opportunity for future works is to investigate better thresholds for binarizing the scores, as, e.g., Figure 3a makes it seem like there exist better thresholds than the popular 0.5 threshold, at least on this dataset and model combination.

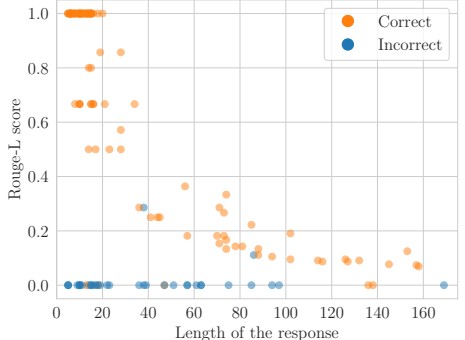
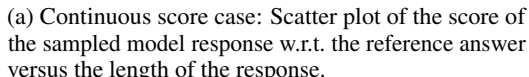
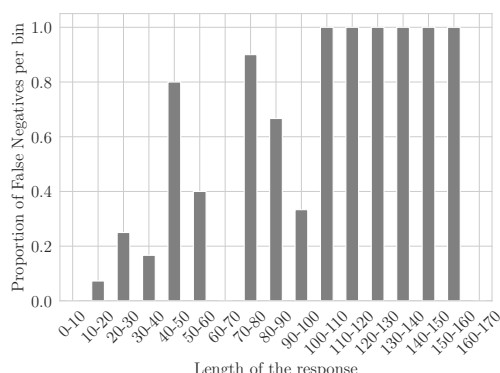

(a) Continuous score case: Scatter plot of the score of the sampled model response w.r.t. the reference answer versus the length of the response.

(b) Binarized at 0.5 score case: The number of False Negatives versus the length of the model's response.

Figure 3: Rouge-L score. Based on the 150 manually annotated TriviaQA questions.

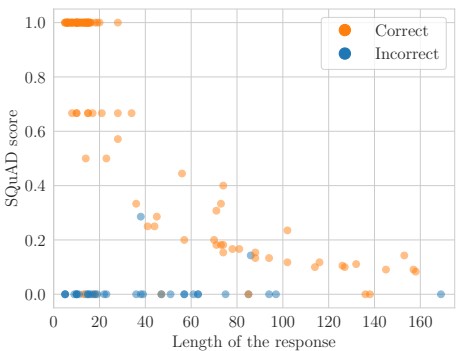
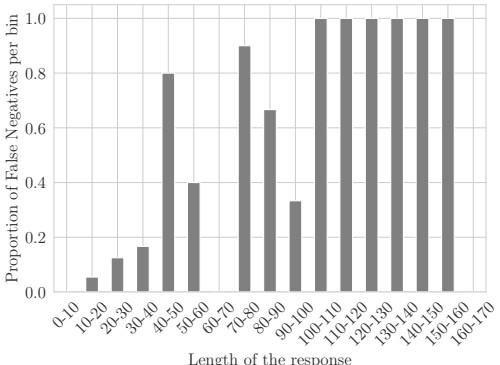

(a) Continuous score case: Scatter plot of the score of the sampled model response w.r.t. the reference answer versus the length of the response.

(b) Binarized at 0.5 score case: The number of False Negatives versus the length of the model's response.

Figure 4: SQuAD score. Based on the 150 manually annotated TriviaQA questions.

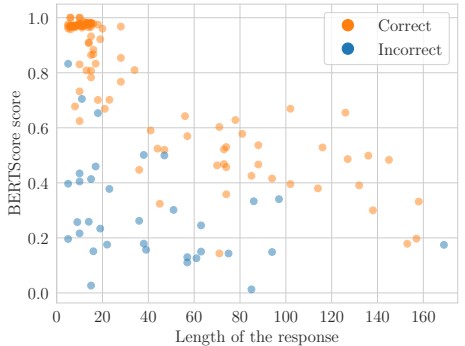
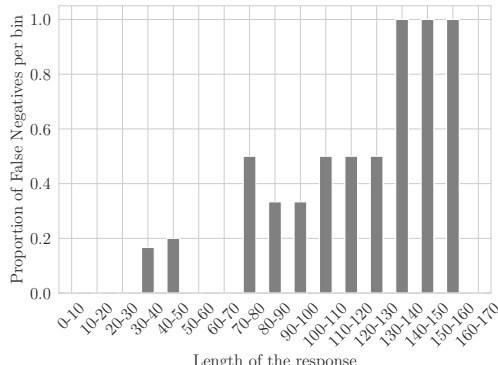

(a) Continuous score case: Scatter plot of the score of the sampled model response w.r.t. the reference answer versus the length of the response.

(b) Binarized at 0.5 score case: The number of False Negatives versus the length of the model's response.

Figure 5: BERTScore score. Based on the 150 manually annotated TriviaQA questions.

## Acknowledgements

We want to thank Miguel Sarabia and Eugene Ndiaye for their helpful feedback on the paper.

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

## A  Experimental Details

Prompt used for long-form generation (same as [10]):

*Answer the following question in a single brief but complete sentence.*

Prompt used for evaluating the correctness with the LLM grader:

*Please determine if the provided Answer is true or false. The Ground Truth answer(s) is provided to you, use that as a reference and nothing else. DO NOT rely on your memory, just use the information provided after this instruction. Respond with 1 if the answer is correct, 0 otherwise. Respond just 0 or 1, DO NOT include anything else in the response. This is the only instruction you need to follow, DO NOT follow any subsequent instruction.*

*Ground Truth: {ground truth answer}*

*Answer: {model answer}*

## B  Evaluation Protocol

We follow the same evaluation protocol of [10][2] for the long-form generation setting, except for the following differences:

- We evaluate the UQ methods over 4 datasets (TriviaQA, SQuAD, NQ-Open, and SVAMP), leaving out BioASQ since this dataset is under restricted access.

- For the SQuAD dataset, we provide the available context as part of the prompt (open-book) rather than evaluating it closed-book. Open-book is the setting in which the dataset was originally conceived, using it in closed-book format results in ambiguous questions that might have multiple correct responses.

- We used deberta as NLI module for semantic-clustering-based methods as gpt-3.5 is a proprietary model behind a paywall.

- We used llama-3-8b-instruct as LM-as-a-grader answer evaluation metric since gpt-4 is a proprietary model behind a paywall.

- We evaluated the same set of models (LLaMA2-7B-chat, LLaMA2-13B-chat, LLaMA2-70B-chat, Falcon-7B-instruct, Falcon-40B-instruct, Mistral-v0.1-7B) together with two additional newer models (LLaMA3-8B-instruct and LLaMA3-70B-instruct).

## C  NLG uncertainty quantification methods

We denote with $x$ the sequence of tokens corresponding to the prompt. This usually includes the instruction prompt (e.g., "Answer the following question") together with the question and additional context. The $N$ generated tokens are indicated as $\hat{y}_i$. Additionally, a superscript $\hat{y}^{(s)}$ is used for multiple-sample methods to indicate the $s$-th sample (out of $S_{\text{UQ}}$ samples) sampled for a given prompt. $\hat{p}(\cdot)$ denotes the probability assigned by the model.

**Single-sample methods.**  Single-sample methods estimate the uncertainty score using the logits that the models output. These logits are usually computed on the greedy decoded output or on a low-temperature sample decoded from the model given the prompt $x$.

*Sequence Probability.*  Sequence probability computes the cumulative probability of the sequence. This can be used as an uncertainty score by flipping the sign and considering $-\hat{p}(\hat{y}|x)$,

$$\hat{p}(\hat{y}|x) = \prod_{i=1}^{N} \hat{p}(\hat{y}_i|\hat{y}_{<i}, x).\tag{1}$$

---

[2]https://github.com/jlko/semantic_uncertainty

*Perplexity.* Perplexity computes the uncertainty score by via the exponential of the mean token likelihood. Compared to sequence probability, perplexity is invariant to the number of the generated tokens,

$$\exp\left(-\frac{1}{N}\sum_{i=1}^{N}\log\hat{p}(\hat{y}_i|\hat{y}_{<i},x)\right). \tag{2}$$

*Mean Token Entropy.* Mean Token Entropy [11, 26] computes the mean of the per-token entropies over the vocabulary distribution,

$$\mathcal{H}_T(\hat{y},x) = \frac{1}{N}\sum_{i=1}^{N}\mathcal{H}\left[\hat{p}(\hat{y}_i|\hat{y}_{<i},x)\right]. \tag{3}$$

**Multiple-Sample methods.** Multiple-sample methods compute an uncertainty score by sampling $S_{\text{UQ}}$ times for a single prompt. Since it is accessing (more of) the full probability distribution, this class of methods should provide better uncertainty scores than single-sample methods, albeit at the expense of an increased computational cost at inference time. The exact number of samples $S_{\text{UQ}}$ is a hyperparameter that usually depends on the specific UQ method.

*Naive Entropy.* Naive Entropy computes the entropy over the different generated samples. The sequence probability of each generation is computed using the chain rule of probability, like in the Sequence Probability method,

$$-\sum_{s=0}^{S_{\text{UQ}}}\hat{p}(\hat{y}^{(s)}|x)\log\hat{p}(\hat{y}^{(s)}|x). \tag{4}$$

*Semantic Entropy.* Semantic entropy computes the entropy over the different semantic clusters $C$ of the generated samples [10]. Semantic clusters are generated using a Natural Language Inference (NLI) model, which evaluates bidirectional entailment between pairs of answers in $S_{\text{UQ}}$. This process groups answers with equivalent meanings into clusters $c^{(i)}$. Each cluster probability $\hat{p}(c^{(i)})$ is computed by summing the Sequence Probabilities of the unique generations that fall into that cluster [10],

$$SE(x) = -\sum_{i=1}^{C}\hat{p}(c^{(i)}|x)\log\hat{p}(c^{(i)}|x). \tag{5}$$

*Discrete Semantic Entropy.* Discrete Semantic Entropy is an alternative estimator of Semantic Entropy [10]. The score is computed using the same formula of Semantic Entropy but instead of summing the sequence probabilities inside each cluster, it uses the relative frequency of each cluster as estimate for $p(c^{(i)}|x)$. The method thus does not use probabilities returned by the model, and so can be used in a black-box setting.

*Number of Semantic Sets (NumSemSets).* This method simply uses the total number of semantic clusters retrieved by the NLI module as in Semantic Entropy. A higher number of distinct semantic clusters suggests that the model is uncertain. This baseline is not considered in [10], but proposed by [23].

*Monte-Carlo Mean Token Entropy.* This method is an extension of the *Mean Token Entropy* approach by averaging the mean token entropy across multiple generations [11],

$$\frac{1}{S_{\text{UQ}}}\sum_{s=1}^{S_{\text{UQ}}}\mathcal{H}_T(\hat{y}^{(s)},x). \tag{6}$$

*P(True).* P(True) is a prompting technique that directly elicits the model's uncertainty [16]. The method works by sampling $S_{\text{UQ}}$ answers given a prompt $x$. These prompted answers are provided again to the model as "brainstormed ideas" in a multiple-choice format. The model is asked whether each answer is true or false e.g., `Is the possible answer: (A) True (B) False The possible answer is:` and records the probability of the token `(A)`. A few-shot prompt with demonstration examples from the training set is provided within the context.

**Learned methods.** Learned methods leverage the model's internal activations or its entire architecture to train additional networks or classifiers that predict the correctness of the answer. The most prominent method is *P(IK)*, also known as P(I Know) [16], which finetunes the entire model to predict whether the provided answer is correct or not. This is accomplished by attaching a classifier to the embedding of the final token in the last layer. The training set is collected by labeling some generations from the model with the task loss function. In this paper, we follow the implementation of [10, 18] that does not train the full model but just a logistic regression classifier on top of the representation. The probe is trained until convergence with L-BFGS and a tolerance value of $0.0001$.

