# OpenReview forum: "On a Spurious Interaction between Uncertainty Scores and Answer Evaluation Metrics in Generative QA Tasks"
_NeurIPS.cc/2024/Workshop/SafeGenAi — SafeGenAi Poster_

### Official Review · Reviewer_TizB · 2024-10-09
**Great work identifying discrepancies between evaluations, pushing for greater reproducibility**

**Rating:** 8
**Confidence:** 4

**Review:**

quality
-------

Pros
- The LLMs and datasets are well-chosen
- The critique is both thorough in its breadth, and well-targeted.
- The comparison of Rouge-L and LLM-grader to human annotation (and high inter-human agreement) is clear, reliable and decisive. The identification of length-bias is also satisfying.

Cons
- What I'd want to see, at the end of a critical work like this, is a list of recommendations. The critique is well-performed, but I'm disappointed that after a thorough review of the parameters which can affect an evaluation the only actionable result I come away with (I may have missed something) is "use an LLM grader, not Rouge-L".

clarity
-------

Pros
- The introduction is wonderfully written, and a delight to read.
- The layout is very clear, and the sections well-defined and signposted.

Cons
- The lengthy introduction to the work's formalism (§3) is something I'd expect to see at the start of a thesis, rather than a conference paper. In particular, it's not clear to me that the overview of sampling methods is targeted and succinct, perhaps more signposting would help highlight this section's relevance to the experiments. While part of the work of the paper is to emphasise the breadth of UQ options, details could be moved to the appendix.
- Despite the extensive formalism, no summary detail is given for Rogue-L itself.


originality
----------
Pros
- I know of no similar literature review or analysis of reproducibility


significance
------------

Pros

- It is, as the authors say, of paramount importance to make sense of the conflicting claims of the performance landscape of the various UQ methods in the literature, to enable reproducability and reliability.
- The paper's focus on the discrepancy between [10] and [11] is keenly targeted, and results in a satisfying attribution.
- The recommendation to use LLM graders is clear and supported.

Cons
- I hadn't really heard of Rouge-L before now, so I wouldn't have expected it to be a particularly significant/prevalent foil for LLM-graders.

---

### Official Review · Reviewer_1yPd · 2024-10-10
**Interesting exploration of ROUGE-L's biases for judging correctness for selective prediction, but findings could be more informative.**

**Rating:** 5
**Confidence:** 4

**Review:**

This paper investigates apparent inconsistencies in the findings of 2 papers on uncertainty quantifications in LLMs. Experiments use a number of different methods (e.g., entropy-based ones, sequence probabilities, and learned selection functions) with 7 models of varying sizes. The aim of the exploration is to examine whether the performance of sequence probabilities may be due to other confounding factors.

**Strengths**

- The explorations of the ROUGE-L metric may be useful to the community. Particularly illuminating the favoring of higher scores is informative and could warrant similar explorations with other metrics. This kind of critical examination of metrics also adds to the growing body of work re-evaluating commonly held conclusions of LLMs performance due to metric choices (e.g., https://arxiv.org/abs/2304.15004), but in the context of selective prediction.

- There is a wide breadth of models and uncertainty quantification methods considered in the experiments, which helps makes the results more comprehensive. The methods used vary quite widely and cover many prevalent unceftainty quantificaiton techniques.


**Weaknesses**

- The effectiveness of LLM-as-a-judge has been established (e.g., https://arxiv.org/abs/2306.05685), while the drawbacks of n-gram matching metrics have also been illustrated in prior work (e.g., https://arxiv.org/abs/2303.16634, https://aclanthology.org/J09-4008). While the problem setting may be different, the usage of these metrics in this work is still quite similar. Hence, it is intuitive that the findings may also be similar, given that abstention performance is linked to task performance. It's not entirely clear how informative of a finding this might be to the community.

- In a similar vein to the previous point, the conclusions of [10] and [11] do not appear to be as at odds as this paper purports. To quote from [10]:  "For LLaMA, there is no clear advantage for any of the methods considered." In that same paper, we can see significant variance in the method rankings between ROUGE-L and BERTScore.

- It could be interesting for those trying to learn from this work to see variation across benchmarks. These results could be in the appendix, but it's plausible that changing the task and evaluation set could yield noticeable differences (e.g., https://arxiv.org/abs/2306.08751).

- Some claims could be adjusted or better supported: For lines 39-44, there do not appear to be any explorations of different temperatures or implementation comparisons, nor budget considerations for evaluation sets.